# Deep Learning-Based Ground Target Detection and Tracking for Aerial Photography from UAVs

**Kaipeng Wang, Zhijun Meng * and Zhe Wu**

School of Aeronautic Science and Engineering, Beihang University, Beijing 100191, China; wangkaipeng105@buaa.edu.cn (K.W.); wuzhe@buaa.edu.cn (Z.W.)
* Correspondence: mengzhijun@buaa.edu.cn

**Abstract:** Target detection and tracking can be widely used in military and civilian scenarios. Unmanned aerial vehicles (UAVs) have high maneuverability and strong concealment, thus they are very suitable for using as a platform for ground target detection and tracking. Most of the existing target detection and tracking algorithms are aimed at conventional targets. Because of the small scale and the incomplete details of the targets in the aerial image, it is difficult to apply the conventional algorithms to aerial photography from UAVs. This paper proposes a ground target image detection and tracking algorithm applied to UAVs using a revised deep learning technology. Aiming at the characteristics of ground targets in aerial images, target detection algorithms and target tracking algorithms are improved. The target detection algorithm is improved to detect small targets on the ground. The target tracking algorithm is designed to recover the target after the target is lost. The target detection and tracking algorithm is verified on the aerial dataset.

**Keywords:** UAV aerial photography; target detection; target tracking; deep learning

## 1. Introduction

In recent years, UAVs have been widely used in military reconnaissance, agricultural planting, forest patrol and other fields. Aerial photography images and videos obtained by UAV have high application value due to their wide viewing angle and low occlusion. Identifying, locating, and tracking valuable targets in aerial images or video is an important part of aerial imagery applications.

The characteristics of aerial image are rich content, varied scene, high angle of view and wide field of view. But the region occupied by the target has a small overall length and is easily occluded by the environment. This puts forward higher requirements for the target detection and tracking process.

In the aerial images, the size of the target is small. Compared with the whole image, the number of pixels occupied by the region of target is small. In this case, the traditional features of the target, such as the red, green and blue (RGB) feature, will reduce the ability to describe the target. Convolution neural network use convolution kernel to enlarge the sensory field of each pixel. Through the convolution layer, a low-resolution region can merge the surrounding pixel information. Therefore, aiming at the small target in aerial images, we use the convolution feature to obtain better feature expression ability. In order to simplify the model structure and reduce the model parameters, deep features are extracted using their corresponding convolution networks according to different detection modules.

This paper proposes a real-time detection-tracking framework for tracking a predetermined target in an aerial real-time picture or video sequence. First, target tracking needs to determine the location of the target. We use the target detection algorithm to determine the target in the initial frame. Then, the comparison algorithm is used to determine whether the predetermined target appears in the picture. Finally, the initial position of the predetermined target is sent to the target tracking algorithm and starts the tracking process.

The framework adopts a combination of target detection and target tracking to ensure tracking accuracy and avoid target loss caused by environmental factors such as occlusion. In this paper, Faster R-CNN [1] and YOLOv3 [2] are used as target detection algorithms respectively, and their convolutional layer is used as the deep features of the target tracking module. The ECO [3] visual tracking algorithm is used as target tracking algorithms. We use OTB50 [4] to test the performance of the target tracking algorithm. Finally, some partial fragments of UAV123 dataset [5] and real scenes are used to test the performance of the target detection-tracking framework.

The main contributions of our work are summarized below:

- We use the convolution features extracted by target detection algorithm for the target tracking process. The feature map with high resolution extracted from a shallow convolution layer has abundant spatial information. The feature map with low resolution extracted from a deep convolution layer expresses stronger sematic information. Using convolution features during target tracking improves the effect of target tracking.
- We propose an online target matching strategy based on SIFT [6]. The predetermined target can be selected from the target detection algorithms using this strategy.
- We propose a tracking score as a criterion for tracking failure detection. This criterion can be used to determine whether the tracking process has failed.

## 2. Related Work

### 2.1. Target Detection Algorithms

Target detection is also referred to as target classification detection. Given an image, algorithm is determined whether there is any target instance of multiple predefined categories in the image, and if so, the spatial locations and categories of each instance are returned. Since AlexNet [7] won the ILSVRC-2012 Challenge, the use of convolutional neural networks for target classification has become mainstream. Overall, today's target detectors can be divided into two main types: two-stage inspection frameworks and single-stage inspection frameworks.

The two-stage detection algorithms include a pre-processing step and a target classification step. The purpose of the pre-processing step is to obtain potential locations that may contain targets. The target classification step is used to determine whether the target exists in the potential location. Girshick et al. proposed the Region CNN (R-CNN) algorithm [8]. The algorithm uses a selective search algorithm to create many proposed regions (ROIs) in the image, and converts these regions into fixed-size images which are sent to a pre-trained SVM model to achieve the target classification task. Subsequently, based on the spatial pyramid pooling network model [9], Girshick proposed Fast R-CNN [10] which obtains the features of each ROI on the deep feature map of the original image and sends it to the fully connected layer for classification and localization. This operation significantly reduces the time required for each image inspection. In 2015, Kaiming He proposed Faster-RCNN [1] which used the region proposal network (RPN) instead of the selective search algorithm and shared the convolution layer parameters of the RPN and R-CNN networks to simplify the operation and improve the inference speed.

The single-stage detection (SSD) framework does not have a region proposal step. Instead, the target's bounding box and category are obtained directly in one step. SSD [11] extracts features through a deep convolutional network and directly uses the convolution kernel to achieve target detection. Since the convolutional layer reduces the resolution of the image, the SSD is operated on multiple scales of feature map to detect different scale targets. The YOLO algorithm [12] regards the target detection task as the regression problem of target region prediction and category prediction, and directly obtains the coordinates of the bounding box, the category information and the confidence of the objects using a single neural network.

Although these algorithms can detect preset types of target from an image, they cannot distinguish between multiple targets of the same type.

*2.2. Target Tracking Algorithms*

Giving the initial bounding box of the target in the first frame of the tracking video, a target tracking algorithm automatically estimates the bounding box of the target object in subsequent frames. The target tracking method is divided into two categories: the generative method and the discriminant method.

The generative methods describe the target area as a generative models in the current frame, and search the area fit to the models as the predicted position in the next frame. The well-known generation methods include the Kalman filtering method and mean shift method. For example, the ASMS algorithm [13], using the mean shift method and color histogram features, has achieved good results.

The discriminant method uses the target region as a positive sample and the background as a negative sample to train a classifier. It uses this classifier to find the optimal region as the position of target in the next frame. The most classical discriminant method is the correlation filtering method. MOSSE [14] uses the correlation filtering of single-channel grayscale features to achieve high-speed tracking of 615 FPS. CSK [15] adds dense sampling and kernel methods based on MOSSE, and it achieved higher accuracy. KCF extended the HOG feature using multi-channel gradients based on CSK method to further improve the accuracy. Martin Danelljan extended CSK with multi-channel color feature, which is known as Color Names, and achieved good results. The HOG feature extracts the gray gradient feature of the image, and the CN feature extracts the color feature of the image. Both of them have become the standard for feature extraction in the tracking algorithm. C-COT [16] used the depth feature of multi-layer convolution for the first time and won the championship in the VOT-2016 competition. Although the accuracy of C-COT is very good, its complexity is high and the operation cannot meet the real-time requirements. Based on C-COT, the ECO algorithm proposes acceleration measures from three aspects: model size, training set size and model update strategy. It can achieve high processing speed and accuracy using hand craft feature including HOG and CN features.

C-COT and ECO use a pre-trained convolution neural network to obtain target features. The pre-trained content of the CNN does not match the actual application scenario.

## 3. Detection-Tracking Framework

*3.1. Framework Architecture*

The workflow of the framework is shown in Figure 1. Feature information of the preset target and the video sequence should be provided in the target detection-tracking framework. The target detection algorithm detects the potential target appearing in the current frame and performs feature matching with the preset target. If there is a preset target in the potential target, the target tracking algorithm is initialized using the target's position to perform continuous tracking of the target. If the target is missed during tracking process, the target detection module will be restarted to find the target again.

*3.2. Target Tracking with Deep Features*

In the aerial images, the size of the target is small. Compared with the whole image, the number of pixels occupied by the region of the target is small. In this case, the traditional features of the target, such as RGB feature, will reduce the ability to describe the target.

Convolution neural network use convolution kernel to enlarge the sensory field of each pixel. Through the convolution layer, a low-resolution region can merge the surrounding pixel information. Therefore, aiming at the small target in aerial images, we use the convolution feature to obtain better feature expression ability.

In order to simplify the model structure and reduce the model parameters, deep features are extracted using their corresponding convolution networks according to different detection modules.

The convolution network structure for extracting deep features is shown in Figure 2. For Faster R-CNN, the VGG-16 network is used to extract deep features. The conv1 layer and the conv4 layer in the convolutional layer are selected as the deep feature output, and

we obtain DF1 and DF2. For YOLOv3, the Darknet-53 network is used to extract deep features. We select the 2nd and 27th convolutional layers as the depth features to obtain the DF1 and DF2 features.

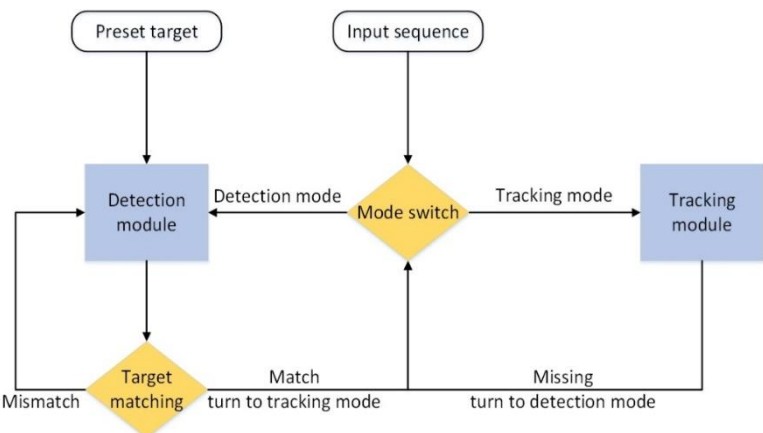

**Figure 1.** Target detection-tracking framework.

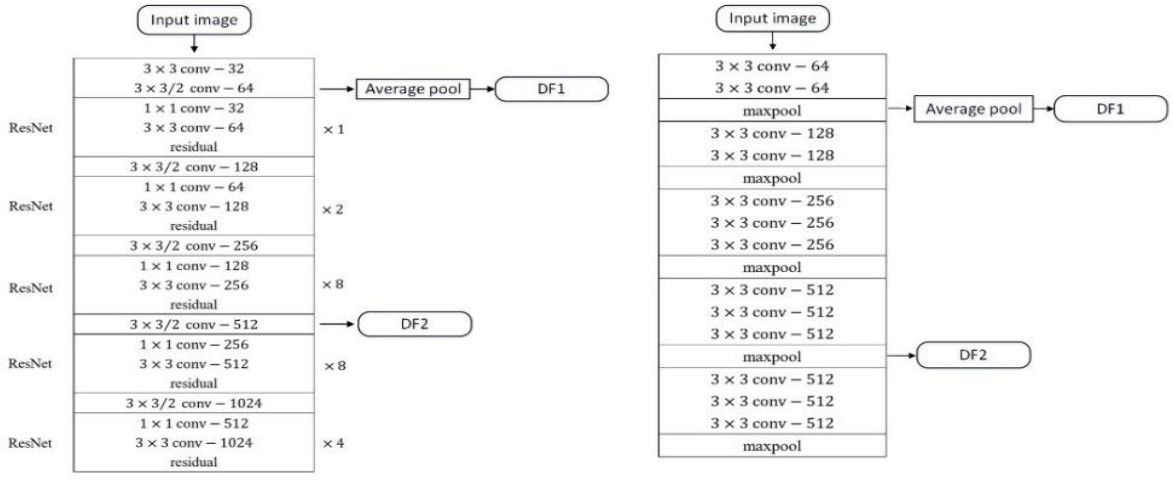

**Figure 2.** Deep feature extracted from CNN. (**a**) corresponds to YOLO v3. (**b**) corresponds to Faster R-CNN.

DF1 obtained from the input image through fewer convolutional layers has a large amount of image spatial information, which represents the low-dimensional image feature of the target itself. DF2 feature has been processed by a large number of convolutional layers and represents the high-dimensional semantic features of the target region.

### 3.3. Target Matching Strategy Based on SIFT

The purpose of target matching is to select the preset target from the potential target extracted from the target detection module. In this paper, the invariant feature transform algorithm (SIFT) is used to calculate features and perform feature matching.

The method to match features is to calculate the Euclidean distance between feature vectors. After the SIFT features of the two images are generated, the feature vectors of image 1 and image 2 are extracted respectively, and the descriptors are calculated. For each feature vector $f$ of the image 1, find two feature vectors $k_1$ and $k_2$ closest to it in the image 2, and calculate the Euclidean distances $d_1$ and $d_2$ between $f$ and $k_1$, $f$ and $k_2$, if $d_1/d_2 < \delta_{SIFT}$ $f$ and $k_1$ are a pair of matching points. An example of the invariant feature

transform algorithm (SIFT) match for the same target with different scenario is shown in Figure 3.

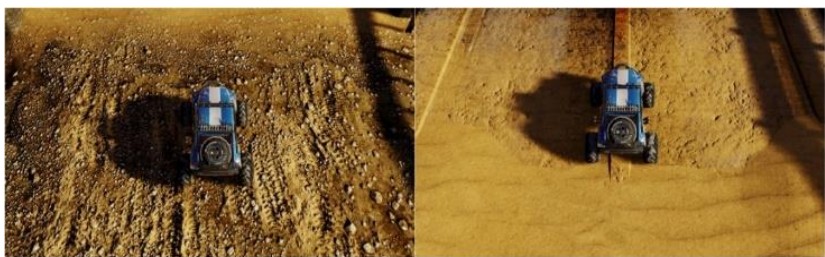

(a) Origin figures

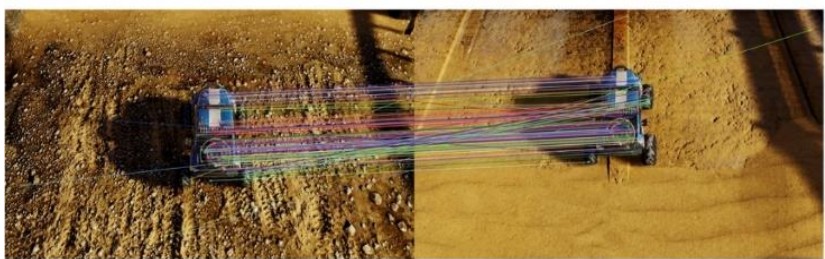

(b) SIFT match result

**Figure 3.** Invariant feature transform algorithm (SIFT) match for the same target with different scenario. (**a**) shows the origin figures. (**b**) shows the SIFT match result for the same target.

Since the conventional SIFT feature calculation is based on grayscale images, color information is lost. But color information in color images is very important. Therefore, it is necessary to improve the SIFT calculation method and use the color information in the image.

Considering that the RGB histogram does not have invariance for the linear changes of gray scale and color, the three channels of RGB are normalized separately. The normalization process is

$$
\begin{pmatrix} R' \\ G' \\ B' \end{pmatrix} = \begin{pmatrix} \frac{R-\mu_R}{\sigma_R} \\ \frac{G-\mu_G}{\sigma_G} \\ \frac{B-\mu_B}{\sigma_B} \end{pmatrix} \tag{1}
$$

where $(R\ G\ B)^T$ and $(R'\ G'B')^T$ are the values of the original and the normalized RGB channels of each pixel, $\mu_*(* = \{R, G, B\})$ and $\sigma_*(* = \{R, G, B\})$ are the mean and variance of each color channel for the entire image respectively.

Each color channel is subtracted from the mean to offset the change in color value, divided by the variance to offset the scale change. The normalized histogram is invariant to the change in color space.

According to the experiment, when the number of matching feature points accounts for more than 50% of the number of key points, we believe that the target matching on the two images is successful.

Target tracking is a continuous process. In this paper, the sample real-time update strategy is used to adapt to the shape and size changes of the target during the tracking process. SIFT features of the target region are calculated and recorded during tracking to achieve the goal of updating the target features in real time. To prevent the sample base from expanding, we delete the previously recorded samples from the sample base and record the new samples. This allows the framework to remember the recent features of the target, facilitates target matching and retrieval, and prevents model drift.

### 3.4. Tracking Score Based on Peak to Sidelobe Ratio (PSR)

MOSSE proposed peak to sidelobe ratio (PSR) to determine tracking failure. MOSSE used an 11 × 11 window around the maximum value in the correlation output as the peak. A similar calculation method is used in this paper.

After calculating the response map $S_f\{x\}$ of all the feature superimpositions, the peak is set to the area of W/16 × H/16 around the maximum response point, and the other areas are set as the sidelobe. The ideal response distribution is shown in Figure 4.

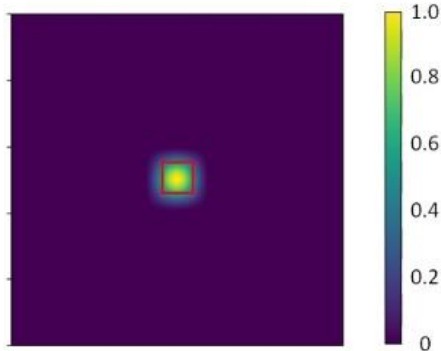

**Figure 4.** Ideal response distribution.

If the filter is correlated with the corresponding target area, target position response is the largest and the sidelobe region response is very small. If the target is disturbed by occlusion, etc., the resulting response map will inevitably destroy this distribution. Therefore the peak sidelobe ratio is calculated as:

$$\text{P} = \frac{S_{max} - \mu_{Sl}}{\sigma_{Sl}} \tag{2}$$

Here, $S_{max}$ is the maximum response of the peak, $\mu_{Sl}$ is the mean of the sidelobe response, and $\sigma_{Sl}$ is the standard deviation of the sidelobe response. When the target area is adapted to the filter, the target can be tracked. At this time, $S_{max}$ is larger. $\mu_{Sl}$ and $\sigma_{Sl}$ are smaller. In this case, the peak to sidelobe ratio P is larger. When the target is occluded, current target area is not adapted to the filter. $S_{max}$ is smaller at this time. $\mu_{Sl}$ and $\sigma_{Sl}$ are larger. In this case, the peak to sidelobe ratio P is smaller. Therefore, whether the target is lost can be judged based on the magnitude of the peak to sidelobe ratio P.

### 4. Evaluation

We use Python to implement target detection and tracking framework. The neural network part uses Tensorflow deep learning library. All the evaluations are tested on a PC with a RTX2060 GPU.

### 4.1. Evaluation of Tracking Module

The tracker with VGG-16 deep feature is called TVGG, and tracker with Darknet-53 deep feature is called "Tdarknet". Two trackers have same parameters except for the depth feature of the target area. Both trackers use FHog feature, IC feature and deep feature. Weight coefficient of FHog feature $\beta_{FHog}$ is 1.0. Weight coefficients of IC feature $\beta_{IC}$ and deep feature $\beta_{feature1}$ $\beta_{feature2}$ is 0.75.

We use OTB50 dataset to test target tracking modules using different deep features. OTB50 has 50 fully annotated datasets to evaluate algorithm performance. It integrates most of the publicly available trackers and sets a unified input and output format for large-scale performance evaluation.

Figure 5 showed the top ten trackers. For a given overlap threshold (image *x*-axis), the success rate (image *y*-axis) represents the proportion of the number of frames in which the tracking frame overlaps with the real frame greater than the overlap threshold in the

total number of frames. The Tdarknet tracker using the darknet deep feature achieves an AUC score of 64.6% with 12 FPS; the TVGG tracker using the VGG deep feature achieve an AUC score of 60.9% with 13 FPS.

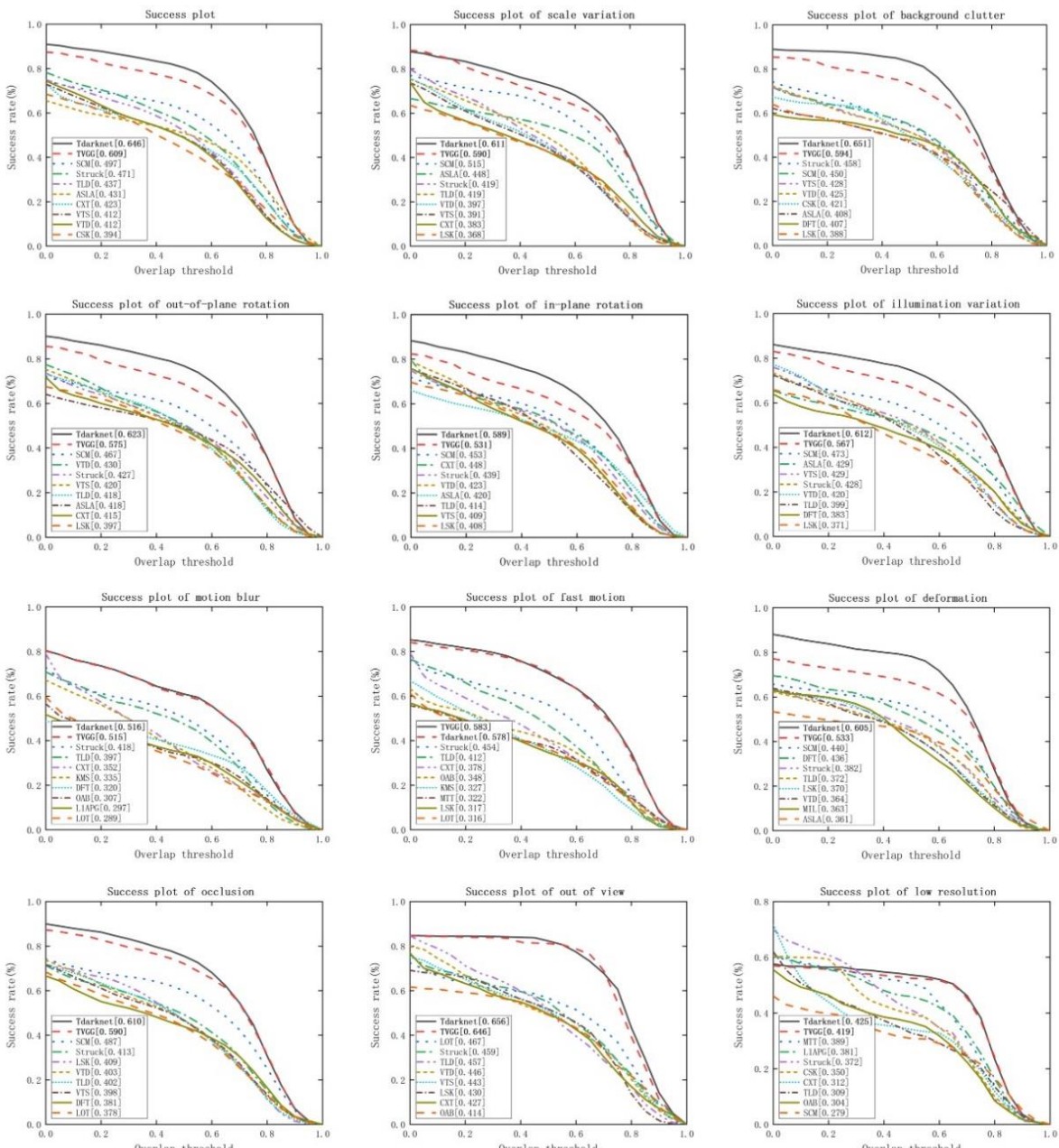

**Figure 5.** Tracking evaluation on OTB50 benchmark. OTB50 has different type of video, include illumination variation, scale variation, occlusion, deformation, motion blur, fast motion, in-plane rotation, out-of-plane rotation, out-of-view, background clutters and low resolution.

We can see that Tdarknet surpasses TVGG in almost all the cases except the "fast motion" case. There are 24 out of 26 videos from the OTB50 dataset "fast motion" case, and the target is very small-scale. We think that the VGG-16 network is sufficient for this scale. On the contrary, the layer structure of the Darknet-53 network is too complicated that too much irrelevant content has been extracted, which interferes with the characteristics of the target itself.

### 4.2. Performance of Aerial Video

The performance of the target detection tracking framework was tested using partial segments of the UAV123 dataset. UAV123 is a new dataset with sequences from an aerial viewpoint. It has a subset of which is meant for long-term aerial tracking.

The car1 sequence in the UAV123 dataset was tested. When calculating the tracking score, the first frame of the tracking process is selected as the standard score $P_S$, The subsequent tracking score $P_C$ is calculated, and the relative score $P_C/P_S$ is calculated. When the relative score is less than 0.3, it is an indication that the object is occluded or tracking has failed, and then target detection is performed again. While framework is initializing, five images are randomly selected from the image in which the target appears in the entire sequence as the initial feature of the SIFT matching algorithm. Initializing the SIFT math algorithm with initial features is shown in Figure 6.

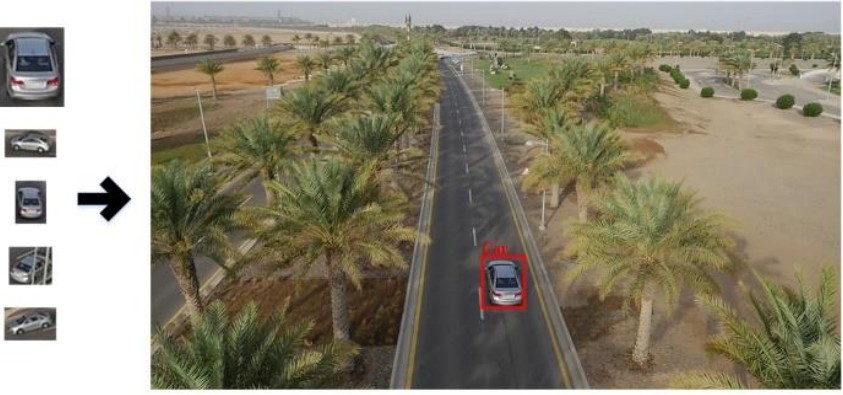

Initial features          First frame

**Figure 6.** Initializing the SIFT math algorithm with initial features.

The results of the framework evaluation using the YOLOv3 detection module and the Tdarknet tracking module is described as follows. The target detection module successfully detects the target in the first frame. The tracking module performs target tracking after the matching. The speed of the frame during the entire detection and tracking process is 11 FPS. The Car1 sequence is a process in which the target vehicle is driving away. In the tracking process of 1800 frames, three target losses occurred.

PSR during the tracking process is shown in Figure 7. Tracking losses occurred at the ABC three points in the figure. The car in A has a large rotation compared with the car in the first frame. Some features in the initial tracking frame are occluded. Therefore, the PSR is less than the threshold. In the next frame after loss, the target detection module successfully found the target back. The car in B has a fast motion during A to B. The PSR is less than the threshold, and the target is successfully retrieved in the next frame after the loss. The car in C has a serious occlusion. After the car is partially blocked, the PSR is less than the threshold. After the car reappears in the field of view, the framework re-detects and retrieves the target again.

The result of the framework test using the Faster R-CNN detection module and the TVGG tracking module is described as follows. The target detection module successfully detects the target in the first frame. The tracking module performs target tracking after the matching. The speed of the frame during the entire inspection and tracking process is 12 FPS.

Relative PSR during the tracking process is shown in Figure 8. The tracking loss occurs at point A. The car in A has a large rotation compared with the car in the first frame. Some features in the initial tracking frame are occluded, so the PSR is less than the threshold and is lost. However, due to the low accuracy of the Faster R-CNN for small targets, the target detection module does not detect this car in the fragment from the 936th to the 1800th frame.

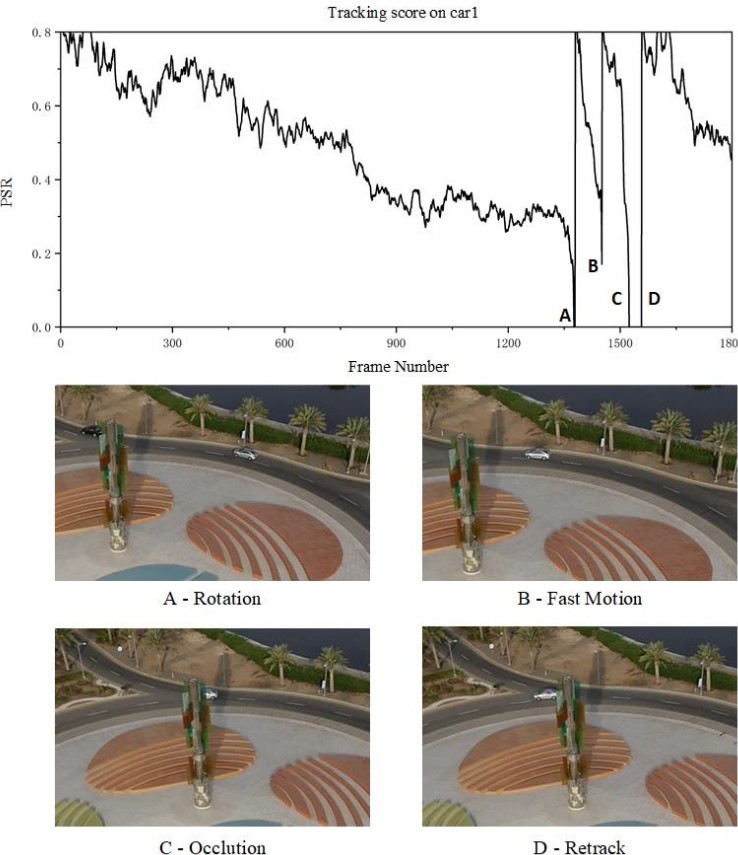

**Figure 7.** Peak to sidelobe ratio (PSR) during tracking process with tracker Tdarknet. This figure shows that the most challenging section of a video can lead to a decrease in PSR.

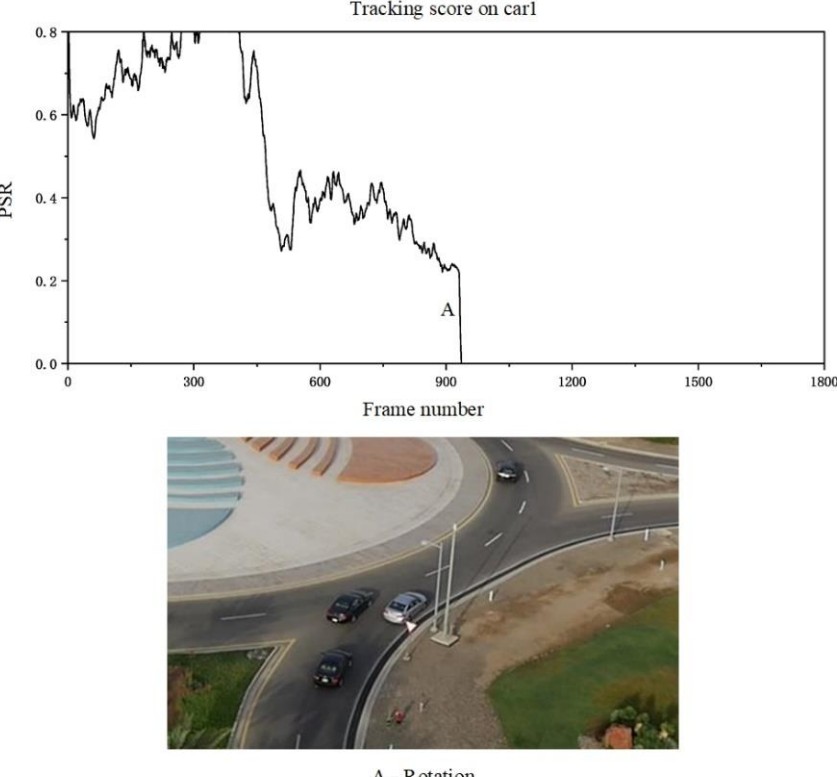

**Figure 8.** PSR during tracking process with tracker TVGG.

It can be seen from the evaluation that target tracking depends on whether the target detection is successful or not. The target in the aerial view scene is far from the camera, and the view angle is large, so that the target takes a small proportion in the image and the target pixel is blurred.

From the comparison of the two detection-tracking frameworks in the above evaluations, we can obtain the following points:

- Faster R-CNN does not detect small targets well, but YOLOv3 is better for small targets in aerial scenes because of multi-scale detection.
- Target tracking is greatly affected by target morphological changes and environmental occlusion. When the target is rotated, it is easy to lose the target because the tracking features are occluded.
- The promoted SIFT matching method provided sufficient target image features during initialization, and collect the current features of the target in the tracking process to ensure that the target can be quickly retrieved after the target is lost.
- The overall performance of the detection tracking framework using the YOLOv3 target detection module and the Tdarknet target tracking module is superior to the detection tracking framework using the Faster R-CNN target detection module and the TVGG target tracking module.

## 5. Conclusions

This paper introduces a renovated workflow of the target detection-tracking framework. Some improvements are proposed for the target detection and target tracking module. We use the convolution features extracted by target detection algorithm for the target tracking process. Using convolution features during target tracking improves the effect of target tracking. We propose a target matching strategy based on SIFT. The predetermined target can be selected from results of the target detection algorithms using this strategy. We propose a tracking score as a criterion for tracking failure detection. This criterion can be used to determine whether the tracking process is failed. The target tracking module is tested using the target tracking dataset. The target detection-tracking framework is tested by aerial videos. Evaluation shows that YOLOv3 has high precision in accomplishing target detection tasks and strong detection ability for small targets. Tdarknet has high accuracy in accomplishing target tracking tasks. However, because of its complexity due to the complexity of the Darknet network, the processing speed is slightly slower.

In fact, we have built a small-scale UAV to undertake the test. The onboard computer is an ARM-core low-cost development board. It can obtain 6~8 frames per second with even very low-resolution images. In the future, we will develop lightweight neural network structure and model quantization skills, so that the detection-tracking architecture could work well in the small-scaled testbed in real time.

**Author Contributions:** Conceptualization, Z.M.; Data curation, Z.M.; Investigation, Z.W.; Methodology, K.W. and Z.M.; Project administration, Z.W.; Software, K.W.; Supervision, Z.W.; Writing—original draft, K.W. All authors have read and agreed to the published version of the manuscript.

**Funding:** This work was supported by the National Natural Science Foundation (NSF) of China (No. 61976014).

**Institutional Review Board Statement:** Not applicable.

**Informed Consent Statement:** Not applicable.

**Data Availability Statement:** All the data in this paper are available on request from the first author.

**Conflicts of Interest:** The authors declare no conflict of interest.

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
