# Peer review of "Deep Learning-Based Ground Target Detection and Tracking for Aerial Photography from UAVs"

_applsci, doi:10.3390/app11188434_

Round 1

Reviewer 1 Report

The paper presents a paper "Deep Learning Based Ground Target Detection and Tracking for Aerial Photography from UAVs". However, I have the following concerns:

  1. The proposed scheme is of great significance to improve detection accuracy. It is recommended to add relevant theoretical analysis to clarify how the deferred decision-making process can accumulate information advantages for subsequent detection, thereby improving detection accuracy.
  2. The motivation of the paper is not clear. Why is your proposal needed? What are the challenges involved? What solutions already exist for the problem you want to solve? What are their limitations and drawbacks?
  3. The results are presented in Fig5. Tracking evaluation on OTB50 benchmark presented are not good. I do not see the information to analyze it.
  4. A practical demonstration of the proposed protocol is not provided.
  5. In the related work section, the authors only point out the problems of different papers and do not summarize the common problems. It is not clear from the related work whether the authors' contributions are unresolved problems in the current research. For instance, there is no information on the drawbacks and limitations of these works. Do the existing works already solve the problem? Why is your proposal needed?
  6. Please discuss the following questions: What solutions already exist for the problem you want to solve? What are their limitations and drawbacks?
  7. One of the major issues of the paper is the level of the English language. The paper includes several typos, grammatical mistakes, and missing punctuation, frequently preventing a complete understanding of the concepts. The authors are required to carefully proof-check their manuscript.

Author Response

Response to Reviewer 1 Comments

Point 1: The proposed scheme is of great significance to improve detection accuracy. It is recommended to add relevant theoretical analysis to clarify how the deferred decision-making process can accumulate information advantages for subsequent detection, thereby improving detection accuracy.

Response 1: Thank you for your advice. The theoretical analysis has been added in the “Introduction” sector. Compared with the whole image, the number of pixels occupied by the region of target is small. In this case, the traditional features of the target will reduce the ability to describe the target. While convolution neural network use convolution kernel to enlarge the sensory field of each pixel. Through the convolution layer, a low-resolution region can merge the surrounding pixel information. Therefore, aiming at the small target in aerial images, we use convolution feature to obtain better feature expression ability.

Point 2: The motivation of the paper is not clear. Why is your proposal needed? What are the challenges involved? What solutions already exist for the problem you want to solve? What are their limitations and drawbacks?

Response 2: The motivation is now clearly clarified in the “Introduction” sector. In the aerial images, the size of the target is small. Compared with the whole image, the number of pixels occupied by the region of target is small. In this case, the traditional features of the target, such as RGB feature, will reduce the ability to describe the target. Convolution neural network use convolution kernel to enlarge the sensory field of each pixel. Through the convolution layer, a low-resolution region can merge the surrounding pixel information. Therefore, aiming at the small target in aerial images, we use convolution feature to obtain better feature expression ability. In order to simplify the model structure and reduce the model parameters, deep features are extracted using their corresponding convolution networks according to different detection modules.

Point 3: The results are presented in Fig5. Tracking evaluation on OTB50 benchmark presented are not good. I do not see the information to analyze it.

Response 3: The deep analysis of the tracking results has been added. Our target detection algorithm is derived from Faster R-CNN which uses VGG-16 network to extract deep features for tracking (marked as TVGG in Fig.5). Ours uses Darknet-53 network to extract deep features for tracking (marked as Tdarknet in Fig.5). We can see that our work surpassed Faster R-CNN in almost all of situations except the “fast motion” case. There are 24 out of 26 videos from the OTB50 dataset “fast motion” case, that the target is very small-scale. We think that the VGG-16 network is enough for this scale, while the layer structure of Darknet-53 network is too complicated that too much irrelevant content is extracted, which interferes with the characteristics of the target itself.

Point 4: A practical demonstration of the proposed protocol is not provided.

Response 4: In fact, we have built a small-scale UAV to do the test. But it’s just a simplified testbed. The image is captured for a USB cam. The resolution is very low compared to real aerial photography. The onboard computer is an ARM-core low-cost development board. It can just get 6~8 frames per second with even low resolution images. And in this paper, our purpose is to improve that the proposed algorithm works well in the open database. We think it’s enough to demonstrate the performing of the algorithm. So the UAV test is not included in the paper. But we are working on some lightweight network that will work fluently on the low-cost UAV platform. It will be published soon.

Point 5: In the related work section, the authors only point out the problems of different papers and do not summarize the common problems. It is not clear from the related work whether the authors' contributions are unresolved problems in the current research. For instance, there is no information on the drawbacks and limitations of these works. Do the existing works already solve the problem? Why is your proposal needed?

Response 5: Sorry, I can’t agree with this comment. We can’t easily find the “drawbacks and limitations” of the previous works of detection and tracking. But it’s about whose work is better in the specific case. The neural network is just a combination of layers and parameters. But some combination works well, while the others not. We learn from the previous works and develop our own detection and tracking structure. And then we evaluate the related works in the same database. The result will show which one is better.

Point 6: Please discuss the following questions: What solutions already exist for the problem you want to solve? What are their limitations and drawbacks?

Response 6: In general, the detection could get mismatch and tracking could fail. We use success rate to evaluate which algorithm is better. For the UAV aerial images, the situation is much more complex, the targets are small-scale and could get occluded. All the previous works can directly work on it. But the success rate would be very low. In this paper, our work surpasses all the previous work in almost all the cases in OTB50 database. We can’t easily say what are the “drawbacks and limitations” of the previous works of detection and tracking. They can work but not good enough compared to the later ones.

Point 7: One of the major issues of the paper is the level of the English language. The paper includes several typos, grammatical mistakes, and missing punctuation, frequently preventing a complete understanding of the concepts. The authors are required to carefully proof-check their manuscript.

Response 7: We have proof-checked the manuscript carefully. And we will get some language editing service from MDPI if necessary.

Thank you very much for your valuable comments.

Reviewer 2 Report

The authors present a novel algorithm in real time for the detection and tracking of targets on the ground used in UAVs, the algorithm presented is based on deep learning. Here some comments

  • the authors should specify the definition of the variables used in equation 1 in a more understandable way
  • How do the authors justify that when the number of matching features points accounts for more than 50% is successful? In line 176
  • Authors should further discuss the contribution of their proposal
  • It is necessary to deepen the discussion of the results obtained in the experiments, discuss their limitations and their areas for improvement.

The authors show a novel methodology for object tracking, the editor would be recommended to accept the paper with minor corrections.

Author Response

Response to Reviewer 2 Comments

Point 1: The authors should specify the definition of the variables used in equation 1 in a more understandable way.

Response 1: The variables in equation 1 are defined just behind the equation in the resubmitted manuscript.

Point 2: How do the authors justify that when the number of matching features points accounts for more than 50% is successful? In line 176.

Response 2: The threshold value 50% is adjustable. We decide this value from a small amount of experiments intuitively, but no quantitative analysis. The threshold value is a balance of  accuracy and success rate. If the value is set too high, we will likely to match the correct target, but miss some targets. On the contrary, if the value is set too low, we will got incorrect matchings. We use this threshold value 50% all over the project. It works well. So it’s decided.

Point 3: Authors should further discuss the contribution of their proposal.

Response 3: The further discussion of our contributions has been added to the “Conclusion” sector.

Point 4: It is necessary to deepen the discussion of the results obtained in the experiments, discuss their limitations and their areas for improvement.

Response 4: The deep analysis of the results has been added after each experiments. And the discussion of our limitations and areas for improvement is discussed in the “Conclusion” sector.

Thank you very much for your valuable comments.

Round 2

Reviewer 1 Report

The paper is good now after the revision.